# Bacteriophage-Based Approach Against Biofilm Infections Associated with Medical Devices: A Narrative Review of ESKAPE Pathogens

**DOI:** 10.3390/ijms26178699

**Published:** 2025-09-06

**Authors:** Karolina Pawłuszkiewicz, Tomasz Busłowicz, Matylda Korgiel, Anita Faltus, Emilia Kucharczyk, Barbara Porębska, Paweł Pochciał, Natalia Kucharczyk, Emil Paluch

**Affiliations:** 1Faculty of Medicine, Wroclaw Medical University, Wybrzeże L. Pasteura 1, 50-367 Wroclaw, Poland; karolina.pawluszkiewicz@student.umw.edu.pl (K.P.); tomasz.buslowicz@student.umw.edu.pl (T.B.); matylda.korgiel@student.umw.edu.pl (M.K.); anita.faltus@student.umw.edu.pl (A.F.); emilia.kucharczyk@student.umw.edu.pl (E.K.); 2Department of Emergency Medicine, Wroclaw Medical University, Borowska 213, 50-556 Wroclaw, Poland; barbara.porebska@umw.edu.pl (B.P.); pawel.pochcial@umw.edu.pl (P.P.); 3Health Care Complex in Oława, ul. Baczyńskiego 1, 55-200 Oława, Poland; kucharczyk.natalia99@gmail.com; 4Department of Microbiology, Faculty of Medicine, Wroclaw Medical University, St. T. Chałubińskiego 4, 50-376 Wroclaw, Poland

**Keywords:** biofilms, bacteriophages, phage therapy, drug resistance, bacterial, health care associated infection

## Abstract

The increasing incidence of hospital-acquired infections and antimicrobial-resistant pathogens poses a major clinical challenge. Nearly all medical devices are vulnerable to bacterial biofilm formation, which acts as a protective coating against the host defense systems and antibiotics. The persistence of biofilm infections, accounting for around 65% of all microbial infections, and poor conventional treatment outcomes has driven interest in alternative approaches like bacteriophage therapy. This review encompasses key aspects of biofilm biology, taking into account the clinically significant ESKAPE pathogens, and provides an in-depth analysis of the role of phage agents in biofilm control as a new biofilm control strategy. Diving deeper into the mechanisms of phage-mediated processes, the review examines how bacteriophages penetrate and disrupt biofilm architecture and evaluates current therapeutic strategies that exploit these actions, acknowledging their limitations and considering possible future directions.

## 1. Introduction

The World Health Organization (WHO) defines a healthcare-associated infection (HAI)—also known as a hospital or nosocomial infection—as an infection acquired during medical care in a hospital or other health facility that was neither present nor incubating at the time of admission. HAIs can emerge in any care setting, even after discharge, and may also affect healthcare workers [1]. The rising number of hospital infections and escalating rates of antimicrobial resistance among pathogens have prompted substantial concern in the medical field. The most recent WHO report states that 15% of patients in low- and middle-income countries and 7% in high-income countries develop at least one HAI during their hospital stay [2]. It is estimated that 60 to 70% of nosocomial infections (NI) are attributable to the use of medical devices [3], however, difficulty in diagnosing device-related infections leads to a likely underestimate of the true prevalence [4]. 

The use of foreign materials—such as catheters for vascular access or urinary drainage, cardiac devices, orthopaedic and dental prostheses, and others—has greatly advanced patient care and clinical outcomes [5]. However, since nearly all devices are susceptible to microbial adhesion and biofilm formation [6,7,8], the inherent risk of infection associated with their use cannot be overlooked. According to the reports of the National Institutes of Health (NIH), about 65% and 80% of microbial and chronic infections, respectively, are caused by microbial biofilms, affecting both tissues and medically implanted devices [9].

A biofilm is a complex, structured community of microorganisms—primarily bacteria—that adhere to living or non-living surfaces and are embedded within a self-produced extracellular polymeric matrix. This adaptive organization includes various bacterial states, such as active, dormant, and persister cells, and provides enhanced protection against environmental stressors and antimicrobial agents [8,10,11]. Among the most commonly identified bacterial species involved in biofilm formation are *Staphylococcus epidermidis*, *Pseudomonas aeruginosa*, *Staphylococcus aureus* (both methicillin-resistant *Staphylococcus aureus* (MRSA) and methicillin-sensitive *Staphylococcus aureus* (MSSA)), *Escherichia coli*, *Klebsiella pneumoniae*, *Proteus mirabilis*, *Acinetobacter baumannii*, *Streptococcus viridans*, *Enterococcus faecalis*, *Enterobacter* spp. [8,12,13]. Notably, several of these species belong to the ESKAPE group of pathogens (*Enterococcus faecium*, *Staphylococcus aureus*, *Klebsiella pneumoniae*, *Acinetobacter baumannii*, *Pseudomonas aeruginosa,* and *Enterobacter* spp.), which are of particular clinical relevance due to their multidrug resistance and prominent role in healthcare-associated infections. The ability of these bacteria to form biofilms further enhances their persistence on medical devices and resistance to antibiotics and host immune responses, thereby contributing significantly to their pathogenicity and treatment failure [14,15]. Colonization is nearly universal upon the devices, if not regularly changed [6,16]. That said, not every device can be easily removed or replaced, particularly in patients who are highly dependent on them. Moreover, in some cases replacement may carry a considerable risk of re-infection on the new device or other unexpected consequences [5,17,18]. 

Biofilms exhibit multifaceted, coordinated defence strategies which lead to increased resistance to therapy and in many cases- its failure [19,20]. The inefficiency in managing infections caused by resistant strains along with the inability to fully eradicate biofilms using conventional antibiotic therapy has driven the search for novel therapeutic strategies [10,13]. Contemporary research is focused on exploring innovative approaches such as antimicrobial peptides (AMPs), bacteriophage therapy, quorum sensing inhibitors, probiotics, physical therapy and others [11,13,21]. Among these, phage therapy has gained growing attention as a novel method for biofilm control due to its unique mode of action against bacterial biofilms. Bacteriophages are viruses that specifically infect bacteria, and they can effectively target biofilm-associated pathogens by both lysing bacterial cells and degrading the surrounding extracellular matrix [22,23].

This review outlines the biological characteristics of biofilms, their underlying resistance mechanisms, and consequent limitations of traditional treatments, with reference to the relevant pathogenic species. It further explores a range of therapeutic approaches in biofilm mediated, device-related infections such as single-phage therapy, phage cocktails, Phage-Antibiotic Synergy (PAS), genetically engineered phages, and the use of phage-derived enzymes, while acknowledging their limitations and considering potential future directions.

## 2. Bacterial Biofilm

### 2.1. Characteristics of Bacterial Biofilm and Biofilm Structure 

Biofilms are structured microbial communities embedded in a self-produced extracellular polymeric substance (EPS) matrix, which acts as a protective barrier. They typically consist of 10–25% microbial cells and 75–90% EPS [9], whose composition varies among the microorganisms involved in biofilm formation.

Table 1 presents a comparative overview of EPS characteristics across the ESKAPE pathogens [24,25,26,27,28,29,30,31,32,33,34,35,36,37,38,39,40,41,42].

Bacteria embedded within the multi-layered EPS of the biofilm are physically separated from the host environment and enter a state of reduced metabolic activity, characterized by decreased nutrient uptake and slower growth. This physiological inactivation allows them to evade the host immune system and contributes to the persistence of the biofilm [43,44]. During biofilm formation, a key structural and functional component is the presence of persister cells. A specialized phenotype of bacterial cells that neither proliferate nor die in the presence of potent antibiotics. These cells, considered dormant variants of the regular bacterial population, play a crucial role in biofilm resilience. Although genetically identical to their parental cells, persister cells exhibit distinct physiological states. Within the biofilm matrix, they can survive exposure to high doses of antibiotics and evade host immune responses. Once the antibiotic pressure subsides, persister cells can reawaken and contribute to repopulating the biofilm, thereby maintaining its stability and chronicity [45,46,47]. Biofilm resistance to antibiotics, disinfectants, and the immune system increases. The body’s immune system tolerates it and does not consider it as a foreign during phagocytosis due to the EPS produced by bacteria [48].

### 2.2. Mechanism of Biofilm Formation 

Biofilm formation is a multi-stage process in which bacteria transition from a free-swimming planktonic state to a sessile community attached to a surface. This transformation is regulated by environmental cues, including temperature, pH, hydrodynamic forces, and the physicochemical nature of the substrate, as well as intracellular signalling molecules [9]. 

Figure 1 presents the process of biofilm formation [9], along with illustrating the differences in matrix EPS composition among the ESKAPE pathogens [24,25,26,27,28,29,30,31,32,33,34,35,36,37,38,39,40,41,42].

Overall, biofilm formation represents a survival strategy that enhances bacterial persistence in hostile environments, particularly on medical devices and indwelling surfaces.

### 2.3. Role of Biofilm in Pathogens

Biofilms protect bacteria from harsh environmental conditions, and their formation is a complex process that begins with the transition of bacteria from a free-swimming planktonic form to a biofilm-producing sessile form [52]. Biofilm formation plays a crucial role in bacterial survival on surfaces, especially on medical devices such as catheters and implants [53]. Nevertheless, biofilm can also form on living surfaces, including bones [54]. It enables bacteria to adhere firmly to these surfaces, form structured communities, and persist in adverse conditions by shielding them from antibiotics and the host immune response [55]. This protective environment promotes chronic and recurrent infections, making bacterial eradication difficult and contributing significantly to increased healthcare-associated infections (HAIs) [32,53].

### 2.4. Risk Factors of Biofilm Formation

Bacterial biofilms are commonly associated with a wide range of medical devices and tissue infections. In clinical settings, biofilm formation on medical devices is a significant concern due to its contribution to persistent infections and treatment resistance. Devices frequently affected include breast implants, ventricular shunts, tissue fillers, ventricular-assist devices, contact lenses, catheters (including urinary catheters), joint prostheses, orthopaedic implants, pacemakers, mechanical heart valves, defibrillators, vascular grafts, endotracheal tubes, and voice prostheses [56].

In addition to device-associated infections, biofilms also play a crucial role in various tissue-related infections, where they contribute to chronicity and poor therapeutic response. Common examples include periodontitis, osteomyelitis, lung infections in cystic fibrosis, endocarditis, dental plaque, chronic tonsillitis, chronic laryngitis, chronic wounds, and biliary and urinary tract infections. These infections often involve biofilm-forming bacteria that persist despite antibiotic therapy and immune defences, complicating patient recovery [57]. 

Table 2 lists the risk factors and conditions that promote biofilm formation in ESKAPE pathogens [9,40,42,56,57].

### 2.5. ESKAPE as Examples of Pathogens Forming Biofilm on Medical Devices 

The ESKAPE group of pathogens (*Enterococcus faecium*, *Staphylococcus aureus*, *Klebsiella pneumoniae*, *Acinetobacter baumannii*, *Pseudomonas aeruginosa*, and *Enterobacter* spp.) has been recognized as a major clinical threat due to its propensity for multidrug resistance and capacity to evade the effects of conventional antimicrobial therapies. In February 2017, these organisms were accorded high-priority status by the World Health Organization (WHO), which highlighted the urgent need for targeted research and the accelerated development of novel antibiotics to address the escalating challenge of antimicrobial resistance [14,15]. Table 3 outlines the WHO 2017 priority classification of ESKAPE pathogens, underscoring their role in antimicrobial resistance and HAIs [14].

ESKAPE pathogens have acquired a broad spectrum of antimicrobial resistance mechanisms through genetic mutations and the horizontal transfer of mobile genetic elements (MGEs) [58]. An important contributor to this persistence is biofilm formation—a complex aggregation of microbial cells embedded within a self-produced extracellular matrix—that limits antibiotic penetration and protects against immune clearance [59]. These mechanisms confer resistance to multiple antibiotic classes, including oxazolidinones, lipopeptides, macrolides, fluoroquinolones, tetracyclines, β-lactams, β-lactam/β-lactamase inhibitor combinations, and last-line agents such as carbapenems, glycopeptides, and polymyxins, despite their clinical limitations [14,15,29,58]. The clinical significance of ESKAPE organisms lies not only in their multidrug resistance but also in their ability to evade host defenses and persist within the host.

Table 4 presents a comparative overview of biofilm-forming characteristics of the ESKAPE pathogens—Staphylococcus aureus, Pseudomonas aeruginosa, Klebsiella pneumoniae, Enterococcus faecium, Acinetobacter baumannii and Enterobacter spp.—specifically in the context of medical device-associated infections [33,34,35,36,37,38,39,40,41,42,60,61,62,63,64,65,66,67,68,69,70,71,72,73,74,75,76]. This synthesis reveals the different mechanisms utilized by each pathogen to establish and sustain their biofilms, thereby underscoring how complex such communities can be when targeted by regular antimicrobial therapies. These mechanistic differences also have implications for bacteriophage therapy design and application.

## 3. Bacteriophages as a Tool in Biofilm Control

### 3.1. Mechanism of Bacteriophage Action, Including Phages’ Ability to Penetrate and Degrade Biofilms

Bacteriophages (phages) have emerged as powerful agents in combating bacterial biofilms–complex, surface-attached microbial communities that are highly resistant to antibiotics [77]. Their efficacy lies in two complementary strategies: direct infection of biofilm-embedded bacteria and degradation of the protective EPS matrix [22,23].

Phages employ various mechanisms to infect bacteria. They can bind to specific bacterial surface receptors, inject their genome, seize control of host cellular machinery to replicate, and produce holins and endolysins. Holins perforate the bacterial membrane, allowing endolysins to access and degrade the peptidoglycan layer causing cell lysis and the release of new virions to infect neighbouring cells within the biofilm [77,78]. Recombinant endolysins that are derived from bacteriophages can effectively disrupt established biofilms even without intact phage particles. For example, LysSYL derived from *Staphylococcus* can remove over 90% of biofilm mass, whereas LysPA26 can work effectively against *P. aeruginosa* biofilms, significantly reducing their biomass [79]. On the other hand, many phages express depolymerases. They are the enzymes that cleave structural polysaccharides and other components of the EPS, such as alginate, capsule, and cellulose. This enzymatic activity reduces biofilm integrity and enhances phage penetration into deeper layers [80]. 

Figure 2 presents the above-discussed mechanisms of phage action, including a short comparison between depolymerase, holin and endolysin characteristic [81,82].

The combined effect of bacterial cell lysis and depolymerase activity results in lysis-matrix synergy. This allows not only the collapse of biofilm structure, but also improved accessibility for immune cells and antibiotics [78,83].

Lastly, bacteriophages’ enzymes also show efficient activity against persister cells, which are typically antibiotic-tolerant. In vitro studies indicate near-complete eradication of persisters following endolysin treatment [79,84].

### 3.2. Different Approaches to Using Phages for Biofilm Elimination

Figure 3 provides a schematic representation of the principal approaches utilized in phage therapy targeting bacterial biofilms.

#### 3.2.1. Phage-Derived Lytic Enzymes

Among the most important compounds in combating biofilms from the heterogeneous group of phage-derived enzymes are endolysins, depolymerases, and VAPGHs (Virion-associated peptidoglycan hydrolases). Endolysins and VAPGHs are compounds that degrade the peptidoglycan of bacterial cell walls, which translates into significant antimicrobial activity against Gram-positive bacteria and reduced activity against Gram-negative pathogens due to the presence of an outer membrane (OM). Depolymerases target the degradation of capsular polysaccharides, lipopolysaccharides, O-polysaccharides, and exopolysaccharides forming the bacterial biofilm matrix [85,86].

The development of molecular techniques has enabled the production of recombinant forms of the above-mentioned enzymes, which have potential for use in therapeutic processes related to the degradation or prevention of bacterial biofilm formation. Recent studies provide many examples of potential applications of enzymes targeting peptidoglycan breakdown. In vivo studies on the endolysin LysSyl (50 µg/mL) demonstrated that MRSA biofilm degradation with its use is possible within just 1–5 h. Additionally, this endolysin shows high efficacy against species-heterogeneous biofilms containing a component of *S. aureus*. LysSyl also exhibited a bactericidal effect against *S. aureus* persister forms, which was enhanced when combined with vancomycin [83]. Another promising example is the endolysin Ply113 (at different concentrations ≥ 8 µg/mL), which in in vitro studies proved effective in combating vancomycin-resistant *Enterococcus faecalis* and vancomycin-resistant *E. faecium*, as well as MRSA strains. A significant effect was also observed in the degradation of mono- and dual-species biofilms involving the above pathogens in various combinations [87]. Other examples of endolysins that could potentially be applied in treating biofilm-forming infections include PM-477 (studied at 16 µg/mL concetration), particularly active against *Gardnerella vaginalis*, and LysCP28 (at 18.7 µg/mL concentration), strongly degrading *Clostridium perfringens* biofilm [88,89]. Recombinant endolysins are also characterized by rapid action, enabling effective degradation of biofilm associated with medical devices in a short time after application (up to 93.5% biofilm mass reduction within 45 minutes by endolysin ClyF at concentration of 50 µg/mL in an in vitro model; complete biofilm degradation after 1 h by 0.25–1 µg/mL enzyme CF-301 and after 4 h by 100 µg/mL CHAPk preparation) [84].

Depolymerases have also been the subject of numerous studies due to their potential in combating biofilms of key bacterial species and strains. Among this group of enzymes is Dpo7, which reduced biofilm mass by 53–85% in two-thirds of tested *S. aureus* strains with pre-treatment of polystyrene surfaces and prevented the formation of these structures. Additionally, Dpo7 (0.15 µM dose) eliminated 90% of cells anchored in the biofilm structure [90]. Another important member of the phage depolymerase group is the TSP enzyme derived from phage ΦAB6. Unlike Dpo7, which widely degrades the EPS, TSP targets more specifically capsular polysaccharides. In studies, TSP showed effectiveness in inhibiting the formation and degradation of *A. baumannii* biofilms, including on Foley catheter surfaces (100 ng dose after 4 h) [91]. Research also highlights the high potential of the depolymerase Dep42, a recombinant protein of the *ORF42* gene isolated from phage SH-KP152226. Dep42 depolymerizes the K47 capsule of *K. pneumoniae*, a multidrug-resistant strain. In addition to its effect on the capsular polysaccharide, Dep42 at concentration of 10 µg/mL also demonstrates high efficacy in degrading *K. pneumoniae* biofilms and inhibiting their formation, particularly when combined with polymyxin [92].

However, it should be noted that biofilm can form independently of polysaccharide structures, which may result in reduced efficacy of depolymerases. An example of a pathogen producing this type of biofilm matrix is *S. aureus* strain V329 [90,93].

#### 3.2.2. Single-Phage Therapy

The term *single-phage therapy* refers to the use of a highly specific type of bacteriophage targeted at a particular strain of a given pathogen, as well as the biofilm structure it forms. In the literature, the term *phage monotherapy* can also be encountered; however, this expression may refer to a therapeutic process in which only bacteriophages are used, without the addition of other therapeutic tools, regardless of whether a single strain or a phage cocktail is applied. A more precise equivalent of *single-phage therapy* is *mono-phage therapy*.

The production process of bacteriophages intended for single-phage therapy begins with the isolation of viral particles from natural environments (sediments, clinical settings, and sewage). Isolation occurs through the incubation of the collected material with a culture of a specific bacterial strain, resulting in the propagation of highly specific phage variants [94,95,96]. The next step is the assessment of the activity of the previously isolated phages [97]. After confirming activity, the main amplification of bacteriophages takes place in cell cultures, followed by filtration and chromatography aimed at removing bacterial residues and their metabolic products, including toxins [98]. The final stage of production includes the evaluation of the quality and safety of the preparation through the assessment of its stability, quality, and immunogenicity [99,100]. 

Phage-based therapies are characterized by high specificity towards particular bacterial species or strains. Numerous studies confirm that even the use of narrow-spectrum phage cocktails (even in addition to systemic antibiotics) avoids interactions between bacteriophages and the patient’s natural microbiota [101,102,103,104]. This highlights the high potential of single phage therapy and narrow-spectrum phage cocktails for safe application in environments rich in commensal microbiota, as well as for the treatment of various infections, including those involving biofilm formation, with minimal risk of dysbiosis [103]. Therapeutic bacteriophages also possess characteristics typical of other viruses, such as the potential for spontaneous or induced changes in their genetic material. The risk of spontaneous alterations in the phage genome has not been excluded [105,106]. The use of single-phage therapy appears particularly beneficial in minimizing the risk of genetic changes leading to the emergence of undesirable phage variants, since the likelihood of recombination is associated with presence of more than one phage type in a therapeutic mixture (*phage cocktail*) [107].

While phage therapy offers targeted action, its high specificity also underlies significant limitations. This form of treatment has limited applicability in cases of heterogeneous infections or biofilms, as well as when the targeted pathogen tends to exhibit high genetic variability. In such situations, in addition to the possible lack of therapeutic efficacy, resistance to *mono-phage therapy* may develop, which is a well-documented phenomenon that can complicate long-term treatment [66,108,109].

The high clinical potential of single-phage therapy in inhibiting biofilm formation and promoting its degradation is supported by numerous findings in current scientific literature. For instance, phage phT4A, used at concentration necessary to achieve an MOI (Multiplicity of Infection) of 10 in a study to assess its effectiveness in degrading and preventing the formation of *Escherichia coli* biofilm, achieved a reduction of 5.5 and 4 log CFU/cm^2^ after 6 h of incubation on plastic and stainless-steel surfaces, respectively. Additionally, phT4A proved effective in preventing biofilm formation [110]. Similarly, phage ZCKP1, administered at 4 h intervals, showed significant reduction of biofilm formed by MDR *K. pneumoniae* KP/01 and prevented the restoration of cell viability for 24 h (best effect at MOI = 50) [100]. Another example of a phage effective in anti-biofilm therapy against *K. pneumoniae*, including MDR strains, is TSK1, which reduced 85–100% of biofilm biomass depending on its age and showed >99% efficacy in preventing the formation of biofilm structure during a 24 h incubation of the pathogen [111]. The literature also reports other examples of specific phage strains effective in monotherapy against biofilms of various pathogens, such as *P. aeruginosa*, *E. faecalis*, and *A. baumanii* [112,113,114,115,116].

#### 3.2.3. Phage Cocktails

The term *phage cocktail* refers to a mixture of bacteriophages that differ in specificity, which may pertain to different strains of the same pathogen or to different bacterial species. The primary advantage of such an approach is the broadened spectrum of activity of the preparation and its efficacy in infections that are heterogeneous at the strain or species level—and consequently, in the biofilms formed during such infections [80,117]. Modern approaches to the development of phage cocktails are based on a two-stage process. The first stage involves the bioinformatic characterization of specific bacteriophages, focusing on the analysis of their spectrum of activity. The range of activity is identified based on the assessment of receptors expressed by the phage. After this preliminary characterization, in vitro evaluation is conducted to confirm effectiveness. Based on the results, it is possible to compose a mixture that minimizes the risk of resistance and maximizes the chances of successful therapy [118].

*Phage cocktails* also demonstrate improved penetration of the biofilm structure. The varied specificity of phages, stemming from differences in gene expression, translates into increased expression of depolymerases with different activities, acting on the biofilm’s EPS structure. This allows for effective targeting of a greater number of EPS components, leading to better loosening of the biofilm matrix and facilitating phage penetration. Through this mechanism, access to bacterial cells within the biofilm is enhanced, promoting phage invasion and replication, which further stimulates biofilm degradation [119,120,121].

Preparations in the form of bacteriophage mixtures also exhibit a reduced probability of resistance development by target bacteria. The pathogen must modify multiple receptors simultaneously to become resistant to a phage cocktail, as its individual components are specific to different structures on the bacterial cell surface. As the number of receptors that need to be altered to achieve full resistance increases, so does the likelihood that more than one mutation in the genes encoding these surface structures will be required. With the growing number of necessary mutations, the probability of developing full resistance decreases [122,123]. Resistance limitation is also supported by phenomena such as the *trade-off* effect, where resistance to one phage may lead to increased susceptibility to another. This may be due to a mutation in one receptor that causes upregulation of another receptor that is targeted by a different phage [124]. Additionally, there may be cases where resistance to a phage is accompanied by the acquisition of another trait beneficial from a therapeutic perspective—for example, increased sensitivity to antibiotics, reduced virulence, or impaired ability to form biofilm [125].

Scientific evidence underscores the superiority of phage cocktails over monotherapies. A cocktail composed of lytic phages FS11, FS17, PS6, and PS8 (prepared by mixing equal volumes of all phages at the same concentration of 10^5^ Plaque Forming Units (PFU) per mL), specific against MDR UPEC (Uropathogenic *E. coli*), reduced biofilm biomass by 86.7%, compared to a 50–66% reduction achieved by the individual phages used separately [126]. Similarly, a cocktail composed of four phages exhibited a stronger lytic effect against XDR *P. aeruginosa* than the single phages used individually. In addition, each phage was used in dose at least 10 times lower than in monotherapy. Notably, the 4-phage cocktail demonstrated higher efficacy compared to the 3-phage cocktail, and the 3-phage cocktail was more effective than the 2-phage combination [127]. Another example of the superiority of cocktails over monophage therapy is provided by *Salmonella enteritidis*, where a cocktail composed of phages E4, E15, and E19 in equal proportions at concentration of 10^10^ PFU/mL each more effectively degraded the biofilm structure than the individual phages or antibiotics at various concentrations [104].

In the use of phage cocktails, the greatest risk lies in the increased likelihood of gene exchange between individual phage variants through homologous and/or non-homologous recombination [128,129]. The most significant modifications in the context of phage therapy are those occurring in the genes of the *hot adaptation site* region, which encode tail proteins. An alternative translation product of the recombined genes may significantly affect the functionality of the phages, and thus the overall efficacy of the phage cocktail therapy. The result of such an event may be the emergence of a new, unpredictable or non-functional phage, for example with an altered or reduced spectrum of activity, disrupting the synergy of the cocktail [105,130,131].

There are commercially available phage preparations being used in clinical practice, particularly in countries like Georgia and Russia, where standardized phage cocktails such as *Intestiphage* and *Pyophage* are accessible over the counter. However, these formulations are not recognized by Western regulatory authorities, limiting their international legitimacy and export. Though phage products are classified as pharmaceuticals in both countries, personalized phage therapy is available only in Georgia. Georgia supports the magistral preparation of personalized phages in specially licensed pharmacies, while Russia prohibits personalized treatments and restricts production to government-authorized cocktails produced by NPO Mikrogen. Both countries face regulatory and scientific limitations, including a lack of double-blinded clinical trials, which hinder global acceptance and further development of their phage therapy programs [132].

#### 3.2.4. Phage and Antibiotic Combination

The combination of bacteriophage preparations with an antibiotic in the treatment process yields a better result than the sum of their separate effects. This phenomenon is referred to as Phage-Antibiotic Synergy (PAS).

The enhanced efficacy is based on a mechanism in which the bacteriophage loosens the biofilm matrix by degrading EPS, allowing freer antibiotic penetration and access to bacterial cells [80]. The antibiotic, on the other hand, stimulates bacterial cell filamentation—that is, inhibition of cell division along with a morphological change in the form of elongation, thereby increasing the cell surface area. Several scientifically proven mechanisms underlie this interaction. The first assumes inhibition of Penicillin-Binding Protein 3 (PBP-3) by β-lactam antibiotics, which is involved in the formation of the septal wall during cell division [133,134]. Antibiotics such as fluoroquinolones (as well as representatives of other classes of therapeutic compounds—e.g., mitomycin C) stimulate cellular mechanisms known as the SOS response, which is a form of reaction to genetic material damage. One of the effects of the SOS response is the inhibition of FtsZ protein expression, which disrupts the formation of the Z-ring—a key structure required for bacterial cell division at the septum [135,136]. A study published in 2006 also identifies amikacin as an antibiotic capable of disrupting Z-ring formation by impairing the translation of proteins, possibly including FtsZ [137]. This suggests the possibility of filamentation following amikacin administration. The increase in cell surface area, along with inhibition of division, causes an increase in effective Multiplicity of Infection (eMOI), that is, the ratio of phage invasion events per one bacterial cell [138]. Another effect of filamentation is also the increase in burst size, that is, the number of phages released per lysis of a single infected bacterial cell. For example, studies indicate that after ciprofloxacin and cephalexin administration, this indicator may increase by 28–36% [139]. Such alteration of eMOI and burst size result in an increased number of phages that can directly participate in biofilm degradation.

Other important properties of PAS from the therapeutic perspective include reduction of the Minimum Inhibitory Concentration (MIC), application against MDR pathogens and the biofilms they produce, as well as inhibition of resistance development [140,141,142,143].

Research demonstrates this synergy in multiple models. The use of a cocktail composed of Klebsiella-targeting phages KPKp and KSKp (MOI = 0.001 for each phage), in combination with ciprofloxacin at sub-MICs (0.5 µg/mL), resulted in approximately a 93.4% reduction in the biofilm mass of MDR *K. pneumoniae* ATCC 700603, compared to approximately 71.4% when using the phage cocktail alone (at MOI = 10) [143]. In a study on novel strategies for combating *P. aeruginosa* biofilm, the phage PhiLCL12, applied concurrently with imipenem at ½ MIC, led to the degradation of nearly the entire biofilm structure after 18 h of treatment [144]. Strategies for combating the biofilm of this pathogen have also been evaluated for their potential applicability in medical device-associated infections. The combination of phage PSP30 with ciprofloxacin demonstrated significant efficacy in eradicating *P. aeruginosa* biofilm grown on titanium coupons, simulating orthopaedic implants [145]. The effectiveness of PAS has also been demonstrated in the treatment of biofilms formed by other pathogens such as *S. aureus* (including MRSA), *E. faecium*, and *A. baumannii* [146,147,148].

#### 3.2.5. Genetically Engineered Phages

The effectiveness of phage-based therapy may be achieved through their genetic modification. The most significant interventions in phage genetic material from the perspective of therapeutic efficacy are those leading to changes in enzyme expression. To enhance the direct impact on biofilm structure, a gene encoding the depolymerase dspB, which is responsible for the hydrolysis of β-1,6-N-acetyl-D-glucosamine—a key adhesin in the formation and maintenance of biofilm integrity in *E. coli* and *Staphylococcus* can be implemented. This modification resulted in a 99.997% biofilm elimination rate in *E. coli* [149].

It is also possible to introduce mechanisms that disrupt quorum sensing (QS), a form of paracrine bacterial communication involved in the development of virulence and biofilm structure formation. QS functionality was disrupted by introducing genome genes into the phage encoding the lactonase Ssopox-W263I, which impairs the above-mentioned type of signaling by degrading acyl-homoserine lactones. This type of modification proved effective in inhibiting biofilm formation by *P. aeruginosa* and *E. coli* [149,150].

Beyond engineering aimed at enzyme expression, modifications to phage character are also known, involving the silencing/removal of genes responsible for the dominance of the lysogenic cycle with the formation of a prophage and a shift toward the lytic cycle, which is desirable from a therapeutic point of view. An example of such genetic intervention was the modified phage ΦEf11, which effectively degraded *E. faecalis* biofilm at a concentration of 5.8 × 10^9^ PFU/mL [121,151]. The shift from lysogenic to lytic character also reduces the risk of horizontal transfer of bacterial genes between cells—including those responsible for virulence or resistance. However, it should be noted that excessive lytic activity of a bacteriophage may lead to reduced therapeutic efficacy, and in some cases decreasing this activity through genetic engineering may be beneficial for therapeutic applications [152].

Other positive effects of genetic engineering of bacteriophages include host range modifications, which involve altering or expanding the spectrum of target pathogens. This process is based on the manipulation of material coding receptor binding proteins (RBP) [153]. Host range modifications may also be performed without genetic material interference. For this purpose, phages are repeatedly incubated with various target strains. As a result, the preparation achieves efficacy in penetrating and degrading the biofilm of the desired pathogen—including MDR strains [154].

Table 5 investigates and compares the current strategies employing bacteriophages for the eradication of bacterial biofilms. 

### 3.3. Clinical Applications of Phage Therapy and Its Relevance to ESKAPE-Related Infections

Figure 4 summarizes selected clinical indications for bacteriophage therapy based on ongoing and registered interventional trials retrieved from ClinicalTrials.gov (accessed on 24 July 2025), using the search term “*Bacteriophage Therapy*.” The conditions represent both localized and systemic bacterial infections where phage-based interventions are being investigated as adjunctive or alternative therapeutics, particularly in the context of multi-drug resistance and biofilm-associated pathology [161].

Although phage therapy is still supported by a relatively limited number of well-designed in vitro and in vivo studies, it has gained increasing attention as a promising alternative to antibiotics—particularly in the treatment of biofilm-associated infections. Carefully structured studies are essential to establish clear guidelines for phage selection, dosing strategies, delivery methods, and to understand long-term outcomes, especially when targeting MDR ESKAPE pathogens embedded in complex biofilm environments. 

Table 6 provides a structured summary of the identified studies, highlighting recent advances in phage-based strategies aimed at eliminating biofilms formed by clinically relevant ESKAPE organisms. A literature review was conducted using PubMed (accessed on 24 July 2025), employing the following search criteria: (Phage-derived lytic enzymes) AND (biofilm) AND (ESKAPE); (Single phage therapy) AND (biofilm) AND (ESKAPE); (Phage cocktails) AND (biofilm) AND (ESKAPE); (PAS) AND (biofilm) AND (ESKAPE); (Engineered phages) AND (biofilm) AND (ESKAPE). The selection was limited to peer-reviewed studies published between 2020 and 2025 [162,163,164,165,166,167]. 

Guła et al. demonstrated that Klebsiella phages (non-specific to *P. aeruginosa: Slopekvirus* KP15, *Drulisvirus* KP34, *Webervirus* KP36), especially KP34, accelerated *P. aeruginosa* PAO1 biofilm formation without affecting planktonic cells. This supports the hypothesis that bacteria can detect phage particles—regardless of specificity—and initiate matrix production as a protective response. Further studies are needed to explore this phage–bacteria cross-talk and its implications for antibiofilm therapy [168].

## 4. Limitations and Implications

Despite very promising results of studies on phage-derived therapies aimed at degradation and prevention of biofilm formation, several significant limitations must still be considered. Monophage therapies are usually characterized by a narrow spectrum of activity, which limits effectiveness against pathogenically heterogeneous biofilms, and increases the likelihood of resistance to the applied preparation [66,108,109]. Phage cocktails, despite their broader spectrum and lower risk of reduced sensitivity to therapy, are not devoid of features hindering clinical application. Their use carries the risk of recombination and thus the acquisition of undesirable traits and/or loss of activity [105,128,129,130,131]. Preparations containing phage-derived enzymes may prove ineffective against biofilms composed of substances that are not substrates of these catalytic proteins, such as extracellular DNA [90,93]. Although therapies using PAS have relatively few problematic features in the clinical context, they remain insufficiently standardized with respect to dosing schemes, timing, and antibiotic selection [158]. In the case of genetically modified phages, the unresolved challenges concern production costs, regulations, and post-modification stability [159,160].

The above-mentioned limitations should constitute the primary direction of further research on the development of phage therapies aimed at combating infections associated with biofilm formation. Solving these issues will be essential for translating laboratory results into reliable clinical practice, and for the popularization of the presented therapeutic strategies on a wide scale.

## 5. Future Perspectives and Research Directions

Despite growing interest in phage-based approaches, there is a notable lack of large-scale, multicenter clinical trials that estimate the efficacy of bacteriophage therapy in the eradication of biofilms associated with medical devices. Table 7 summarizes selected clinical trials registered on ClinicalTrials.gov (accessed on 24 July 2025) related to the application of bacteriophage-based therapeutics against ESKAPE pathogens in the context of medical device–associated infections, such as prosthetic joint infections [169,170,171,172]. Three clinical trials investigating bacteriophage therapy for prosthetic joint infections (PJIs) caused by *S. aureus*, *S. epidermidis*, *Staphylococcus lugdunensis*, *Streptococcus* spp., *Enterococcus faecium*, *E. faecalis*, *E. coli*, *P. aeruginosa*, and/or *K. pneumoniae* were withdrawn prior to completion. Trial NCT04787250 was terminated due to protocol modifications, while NCT05269121 and NCT05269134 were discontinued following sponsor-related decisions [173,174,175].

Bacteriophage therapy offers a compelling and biologically targeted strategy for combating infections caused by ESKAPE pathogens, particularly in clinical scenarios involving antibiotic-resistant biofilms that form on indwelling medical devices. These persistent infections often pose significant therapeutic challenges due to the protective nature of the biofilm matrix and the limited efficacy of conventional antimicrobial agents. By exploiting the natural lytic activity of phages and, in some cases, their ability to degrade extracellular polymeric substances, this approach holds considerable promise for enhancing infection control in high-risk, device-associated settings. 

## 6. Materials and Methods

A comprehensive literature search was conducted using databases such as PubMed, Google Scholar, and Scopus, focusing primarily on studies published between 2020 and 2025; however, earlier pivotal publications were also included when deemed relevant. Additional sources such as Cochrane Library and ClinicalTrials.gov were used to identify grey literature and unpublished data. Core search terms included combinations of: “biofilm”, “medical device”, “HAI”, “biofilm formation”, “antimicrobial resistance”, “bacteriophage”, and “phage therapy”. While advanced search methods prioritized literature from the past five years, this review analyses research spanning from 1999 to 2025. Figure 1, Figure 2, Figure 3 and Figure 4 were created using diagrams.net (formerly draw.io), web app version 14.6.13 (Accessed on 18 July 2025), and Autodesk Sketchbook for iPad, version 6.2 (released 23 May 2025)—all based on the authors’ interpretation of synthesized data from multiple studies.

## 7. Conclusions

Biofilm formation is one of the major survival strategy for bacteria under unfavourable conditions, especially on the surfaces of medical devices. Pathogens commonly associated with biofilms include *Staphylococcus aureus*, *Pseudomonas aeruginosa*, *Klebsiella pneumoniae*, *Staphylococcus epidermidis*, *Enterococcus faecalis*, *Escherichia coli*, *Proteus mirabilis*, and *Acinetobacter baumannii*. Several of these species belong to the ESKAPE group of pathogens and are of particular clinical relevance due to their multidrug resistance and prominent role in healthcare-associated infections. These organisms differ in biofilm composition, antibiotics resistance mechanisms, and adhesion strategies, with some showing preferences for specific medical surfaces. While several approaches have been explored to address this issue, bacteriophage-based strategies have recently garnered growing interest. Monophage therapy, phage cocktails, phage-derived enzymes, PAS and genetically engineered phages all show promise, each with distinct modes of action and potential benefits. However, despite encouraging results in in vitro and in vivo models, strong clinical evidence is still lacking. In this review, we have included both published studies and ongoing or registered clinical trials that address this topic. What becomes clear is that, although the field is developing, the number of large-scale, well-designed clinical investigations remains limited. There is an urgent need for multicenter studies that assess the safety, efficacy, and practical use of phage therapy—particularly in the context of persistent infections involving medical devices and multidrug-resistant strains. In future, collaborative efforts between researchers, clinicians, and regulatory bodies will be essential to translate promising laboratory findings into effective and accessible clinical solutions.

## Figures and Tables

**Figure 1 ijms-26-08699-f001:**
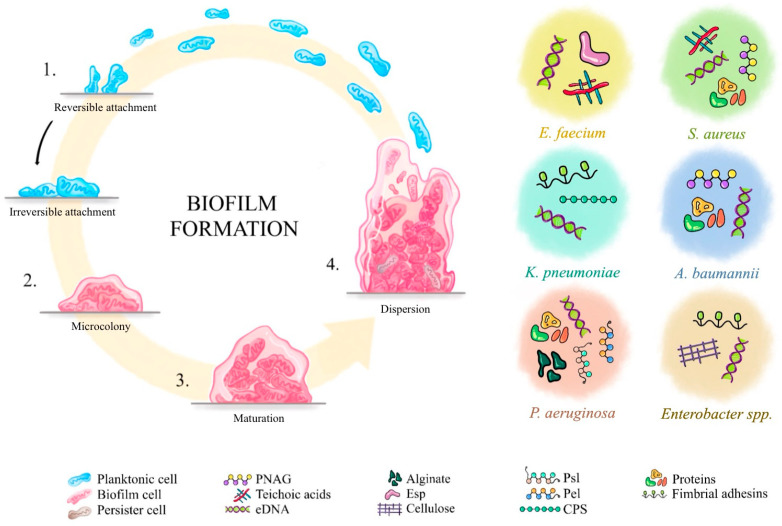
Biofilm formation: Step-by-step process and extracellular polymeric substance (EPS) composition across the ESKAPE group [9,24,25,26,27,28,29,30,31,32,33,34,35,36,37,38,39,40,41,42]. 1. The process begins with initial attachment, where planktonic bacteria reversibly adhere to a surface. Upon surface recognition, the levels of intracellular signalling molecules (e.g., cyclic diguanylate monophosphate (c-di-GMP)) increase, promoting irreversible adhesion and downregulation of motility [30,49]. Surface-sensing systems, such as pili-associated chemotaxis-like system (Pil-Chp), are instrumental in this transition, converting planktonic cells into surface-aware phenotypes. 2. Adherent bacteria proliferate and form microcolonies embedded in a self-produced EPS matrix. Flagella and type IV pili support movement, and cell–cell interactions required for early biofilm organization [50]. 3. As the biofilm matures, it develops a structured three-dimensional architecture. The extracellular matrix ensures mechanical stability, protects against antimicrobials and immune responses, and retains quorum-sensing signals [51]. Mature biofilms often show stratified forms, shaped by gradients of nutrients and oxygen [50,51]. 4. In the final stage, dispersion occurs through active mechanisms—such as increased motility and matrix degradation—or passively via shear stress or nutrient scarcity, allowing colonization of new sites [31]. Abbreviations: PNAG—poly-β-(1,6)-N-acetylglucosamine; eDNA—extracellular DNA; Pel—glucose-rich polysaccharide; Psl—mannose- and galactose-rich polysaccharide; CPS—Capsule polysachharides; Esp—Enterococcal surface protein; *E. faecium*—*Enterococcus faecium*; *S. aureus*—*Staphylococcus aureus*; *K. pneumoniae*—*Klebsiella pneuomiae*; *A. baumannii*—*Acinetobacter baumannii*; *P. aeruginosa*—*Pseudomonas aeruginosa*.

**Figure 2 ijms-26-08699-f002:**
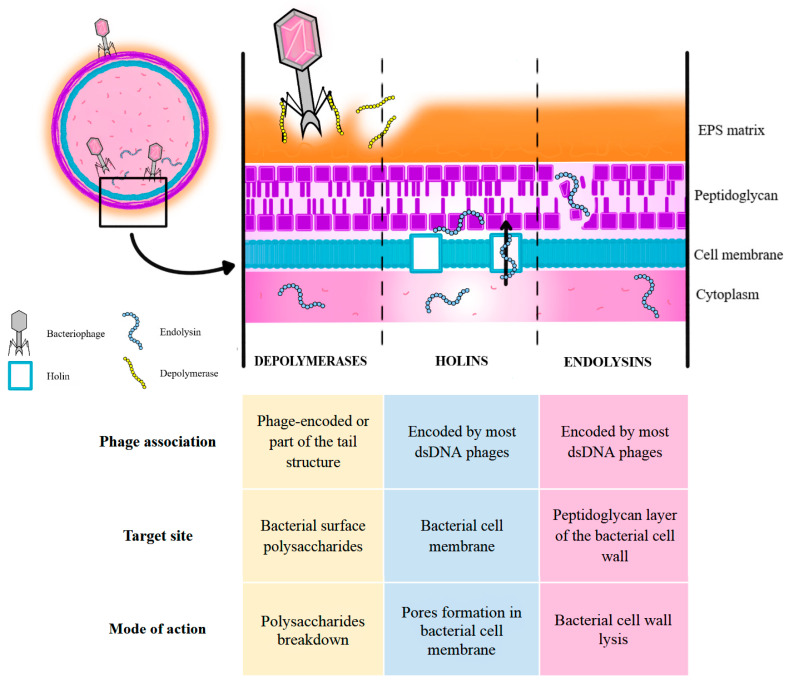
Characteristics of action modes of depolymerases, holins, and endolysins (as an example, in a Gram-positive bacterium) [81,82]. Abbreviations: EPS—extracellular polysaccharidic substance; dsDNA—double-stranded DNA; G(+)/(−)—Gram-positive/-negative.

**Figure 3 ijms-26-08699-f003:**
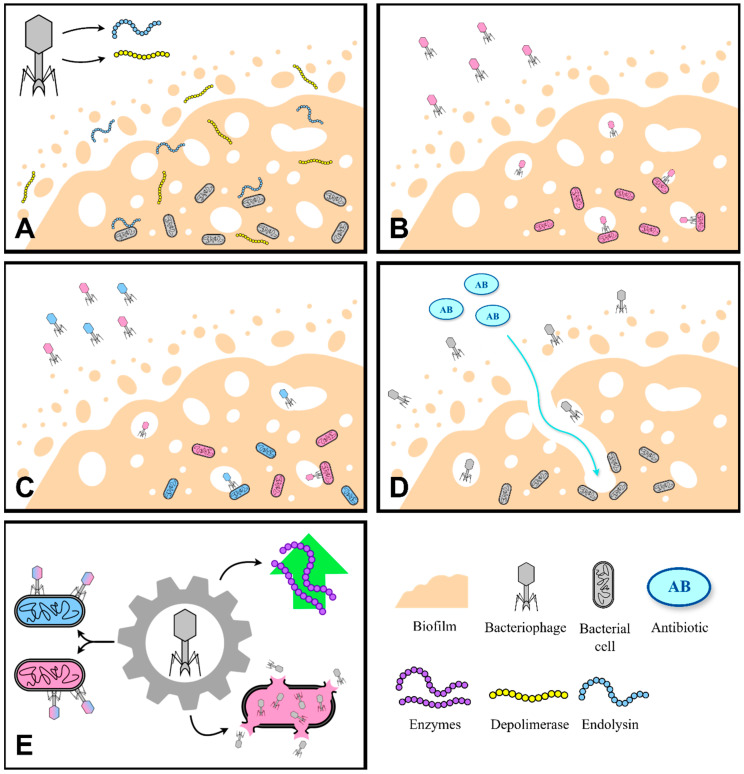
Graphic representation of different approaches in phage therapy against biofilm. (**A**) phage-derived enzymes; (**B**) monophage therapy; (**C**) phage cocktail; (**D**) phage-antibiotic synergy; (**E**) engineered phages.

**Figure 4 ijms-26-08699-f004:**
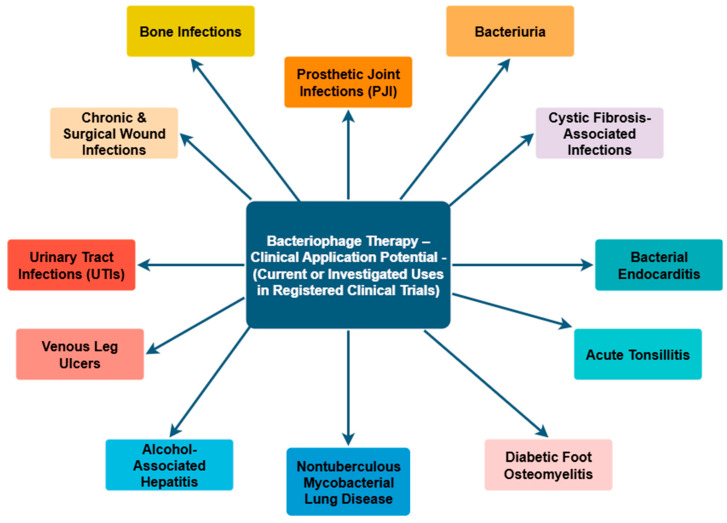
Bacteriophage therapy in practice and research: Insights from current and registered clinical trials [161].

**Table 1 ijms-26-08699-t001:** Comparative EPS characteristics of ESKAPE pathogens [24,25,26,27,28,29,30,31,32,33,34,35,36,37,38,39,40,41,42].

Pathogen	EPS Components	References
*Staphylococcus aureus*	- PIA/PNAG - Teichoic acids - eDNA - Proteins: coagulase, adhesins, proteases	[25,26,27]
*Pseudomonas aeruginosa*	- Alginate - Pel - Psl - eDNA via autolysis - Proteins: lectins LecA, LecB	[24,25]
*Klebsiella pneumoniae*	- CPS - Colanic acid-like EPS - Fimbrial adhesins type 1 & 3 - eDNA	[25,26,27,38,39]
*Enterococcus faecium*	- Polysaccharide antigen - Esp - Lipoteichoic acids - eDNA	[25,26,27,29,32]
*Acinetobacter baumannii*	- PNAG - Capsule polysaccharides - eDNA - Proteins	[25,26,27,29,40]
*Enterobacter* spp.	- Colanic acid-like polysaccharides - Cellulose - Fimbrial adhesins - eDNA	[25,27,41,42]

Abbreviations: PIA—Polysaccharide intercellular adhesin; PNAG—poly-β-(1,6)-N-acetylglucosamine; eDNA—extracellular DNA; Pel—glucose-rich polysaccharide; Psl—mannose- and galactose-rich polysaccharide; CPS—Capsule polysaccharides; Esp—Enterococcal surface protein.

**Table 2 ijms-26-08699-t002:** Risk factors of biofilm formation in ESKAPE pathogens [9,40,42,56,57].

Pathogen	Risk Factors	References
*Staphylococcus aureus*	- Catheters, mechanical heart valves, hip prosthesis - Chronic wounds, severe cutaneous infections, and skin diseases - Protein recycling - Matrix proteins enhancing flexibility, adaptation, and mixed species of biofilm - SCVs—diverse phenotype within biofilms	[9,56]
*Pseudomonas aeruginosa*	- Efflux pumps - OprD oprin - Wounds, cystic fibrosis, and other chronic infections - Endotracheal tubes, contact lenses - Protective response to stress - Neutralizing enzymes (cephalosporinase AmpC) - Slow growth rate inside colony - SCVs—diverse phenotype within biofilms - Matrix acidification	[9,56,57]
*Klebsiella pneumoniae*	- Temperature from 35 °C to 40 °C - Consistent growth on abiotic surfaces - Mixed strains of *Klebsiella penumonaiae* - Infectious urinary stones	[9,56]
*Enterococcus faecium*	- Intestinal infections - Hip prosthesis - Local environment	[56,57]
*Acinetobacter baumannii*	- Temperature, osmolarity, ferrous iron concentration - Nutrients and glucose availability - Ambient acidic conditions - Hydrophobicity and oxygen content	[40]
*Enterobacter* spp.	- SCVs—diverse phenotype within biofilms - Urological catheters - Also risk of biofilm formation on: dental materials, nasogastric and orogastric enteral feeding tubes, prostheses, and other medical devices - Body temperature (37 °C) > room temperature (24 °C)	[9,42]

Abbreviations: SCVs—Small colony variants.

**Table 3 ijms-26-08699-t003:** ESKAPE pathogens designated as critical or high priority by WHO (2017) [14].

Priority	Name of Bacteria from ESKAPE
1: CRITICAL	*Acinetobacter baumannii*, carbapenem-resistant; *Pseudomonas aeruginosa*, carbapenem-resistant; *Enterobacteriaceae*, carbapenem-resistant, 3rd generation cephalosporin-resistant
2: HIGH	*Enterococcus faecium*, vancomycin-resistant; *Staphylococcus aureus*, methicillin-resistant, vancomycin intermediate and -resistant

**Table 4 ijms-26-08699-t004:** Comparative biofilm characteristics of ESKAPE pathogens on medical devices [33,34,35,36,37,38,39,40,41,42,60,61,62,63,64,65,66,67,68,69,70,71,72,73,74,75,76].

Feature	*Staphylococcus aureus*	*Pseudomonas aeruginosa*	*Klebsiella pneumoniae*	*Enterococcus faecium*	*Acinetobacter baumannii*	*Enterobacter* spp.
Gram-stain and shape	G (+) cocci	G (−) rod	G (−) rod	G (+) cocci	G (−) rod	G (−) rod
Antibiotic resistance	MRSA	Carbapenem–resistant *P. aeruginosa*	ESBL	VREs	Carbapenem–resistant *A. baumannii*	ESBL, carbapenem– resistant *Enterobacter* spp.
Adhesion mechanism	Fibronectin binding proteins A, B, clumping factors, A and B, and collagen binding proteins (MSCRAMMs)	Type IV pili, alginate	Type III fimbriae fim (homolog of enterococcal ebp), type I fimbriae, type VI protein secretion system	ESP, MSCRAMMAce, aggregation substance, capsule	Capsule (cell-to-cell adhesion)	Type VI secretion system, enterobactin
Biofilm formation	Aggregation substance	Type IV pili	Capsular polysaccharide, type III fimbriae	Capsule, cell wall polysaccharide, aggregation substance	Capsular polysaccharide	Capsule
Device adhesion	Prosthetic joints, pacemakers, vascular catheters	Ventilator tubing, urinary catheters, central lines	Endotracheal tubes, duodenoscopes, urinary catheters	Central venous catheters, prosthetic valves	Ventilators, central lines, urinary catheters	Urinary catheters, transplant–related devices
Infections	Wound infections, multiple soft tissue infections, infective carditis, bacteremia, fatal pneumonia	Immunocompromised patients, isolated from CF and burn patients. Nosocomial infections–ventilator-associated pneumonia, urinary tract infections, central line bloodstream infections, surgical infections	Community–acquired pneumonia, urinary tract, blood stream, and brain infections	Catheter–associated urinary tract infections, surgical site infections, bloostream infections	Critically ill patients who are severely immunocompromised, hospital-acquired respiratory infections and urinary tract, wound infections	Bacteremia, urinary tract infections, surgical site infections, device-related infections
References	[33,34,35,60,61,63,73]	[36,37,62,63,64,65,66,67,68,69,70,74]	[38,39,71,72,75]	[76]	[40]	[41,42]

Abbreviations: MRSA—Methicillin-resistant *S. aureus*; ESBL—Extended-spectrum b-lactamase; VREs—Vancomycin-resistant enterococci; MSCRAMMs—microbial surface components recognizing adhesive matrix molecules; ESP—Enterococcal surface protein sp.; MSCRAMMAce—collagen binding proteins Ace; CF—cystic fibrosis.

**Table 5 ijms-26-08699-t005:** Comparison of approaches in phage use for biofilm elimination—description of characteristic, effectiveness, advantages, and disadvantages of different approaches [66,80,84,85,86,100,103,104,105,107,108,109,110,111,112,113,114,115,116,117,119,120,121,122,123,124,125,126,127,128,129,130,131,133,134,135,136,137,138,139,140,141,142,143,144,145,149,150,151,155,156,157,158,159,160].

Type of Phage Therapy	Characteristic	Advantages	Disadvantages	Effectiveness (In Vitro/In Vivo)
Phage-derived lytic enzymes	Direct influence on the biofilm matrix components without the use of phage [85,86]	No use of viral vectors, quick action onset, low risk of resistance [84,155,156]	Limited effectiveness on G(−) bacteria [85,86]	High especially on G(+) bacteria [85,86]
Single-phage therapy	Influence on a specific strain of the pathogen and its biofilm [157]	High specificity, low risk of dysbiosis, lower risk of genetic recombination [103,107]	Narrow spectrum, risk of resistance development [66,108,109]	High on specific pathogen strains [100,110,111,112,113,114,115,116]
Phage cocktails	Influence on heterogenic-strain pathogen and its biofilm or multi-species infections and biofilms; better biofilm penetration [80,117,119,120,121]	Wide spectrum, low risk of resistance development [122,123,124,125]	Interphage genetic recombination risk [105,128,129,130,131]	High [104,126,127]
PAS	Increased penetration of antibiotics, increased susceptibility to phages [80,133,134,135,136,137,138,139]	Lower medication doses, effectiveness on MDR biofilms [140,141,142,143]	Lack of standardized medication schematics [158]	High, synergistic effect [143,144,145]
Engineered phages	Increased production of biofilm-degrading enzymes, modifications of lytic/lysogenic character, host range modifications [149]	enhanced biofilm degradation, wider host range	High costs, regulation problems, uncertain genomic stability [159,160]	Very high [121,149,150,151]

Abbreviations: PAS—Phage antibiotic synergy; MDR—multidrug-resistant pathogens; G(+/−)—Gram-positive/negative.

**Table 6 ijms-26-08699-t006:** Summary of recent studies (2020–2025) investigating bacteriophage-based strategies for the elimination of ESKAPE pathogen biofilms [162,163,164,165,166,167].

Target Pathogen	Phage(s) Used	Key Findings	Study Authors and Date
MDR *Klebsiella pneumoniae* biofilm	KP34 (depolymerase-producing), KP15 (non-depolymerase), recombinant KP34p57 enzyme	Phage KP34 achieved a ~3-log biofilm reduction, further enhanced to 4 logs when combined with KP15. The triple combination with ciprofloxacin led to a 5.7-log decrease. While KP34p57 depolymerase alone showed minimal effect, it significantly boosted phage efficacy, highlighting its role as a supportive agent.	Latka et al. (2020) [162]
MDR *K. pneumoniae* biofilm	vB_KpnS_FZ10 vB_KpnS_FZ41, vB_KpnP_FZ12 vB_KpnM_FZ14	Three phages showed halo zones linked to depolymerase activity, helping break down capsules and biofilms. The fourth, vB_KpnS_FZ41, lacked these enzymes and had a narrower host range. A cocktail of all four phages lysed all tested *K. pneumoniae* strains, showing the benefit of combining phages with and without depolymerases to broaden effectiveness.	Zurabov et al. (2021) [163]
*K. pneumoniae* biofilm	A¥L and A¥M which belonged to Myoviridae and Siphoviridae family	When applied individually or in combination, they achieved 50–70% reduction of mature (48 h old) *K. pneumoniae* biofilms in vitro. Significant biofilm disruption and bacterial killing were further confirmed through live/dead fluorescence staining and scanning electron microscopy.	Asghar et al. (2022) [164]
MDR *Acinetobacter baumannii* biofilm	vB_AbaM_ABPW7	Phage vABPW7 significantly reduced biofilm biomass and successfully eradicated preformed biofilms. In an A549 human alveolar epithelial cell model, it effectively decreased both planktonic bacterial load and bacterial adhesion, without inducing any detectable cytotoxic effects.	Wintachai et al. (2022) [165]
MDR and XDR isolates of *A. baumannii*, *K. pneumoniae*, and *Pseudomonas aeruginosa* biofilm	Phage cocktails	From 81 hospital wastewater samples, 31 phages targeting MDR bacteria were isolated. Phage cocktails showed the best results, fully eradicating *A. baumannii* biofilms with colistin at just 1–2 µg/mL. In *P. aeruginosa*, strong phage–antibiotic synergy was observed, with MBECs reduced up to 64-fold. While effects in *K. pneumoniae* were less consistent, two strains also responded better to combined phage–colistin treatment.	Ragupathi et al. (2023) [166]
*P. aeruginosa* strain PAO1 biofilm	PaPC1, PaWP1, and PaWP2	Each phage in the in vitro study significantly reduced 24 h old *P. aeruginosa* biofilms on polystyrene—PaPC1 by 66.7%, PaWP1 by 39.1%, and PaWP2 by 62.9%. When combined, the phage cocktail achieved over 75% reduction, showing clear synergistic effects.	Kovacs et al. (2024) [167]

Abbreviations: MDR—Multidrug-resistant; XDR—Extensively drug-resistant.

**Table 7 ijms-26-08699-t007:** Ongoing clinical trials involving bacteriophage therapy targeting ESKAPE pathogens in medical device–associated infections [169,170,171,172].

ClinicalTrials. Gov ID and Research Status	Pathogen	Conditions	Intervention/Treatmentclassification	Intervention/ Treatment–Informations	Reaserch Phase
NCT06942624 [169]; Not yet recruiting	A methicillin-susceptible *Enterococcus faecium*	PJI of the hip	Biological: Phage Therapy	Lytic phages (saline-magnesium buffer)–intravenous and intra-articular administration, twice daily for 14 days; combined with standard antibiotic therapy	I, II
NCT06456424 [170]; Active, not recruiting	A methicillin-susceptible *Staphylococcus aureus*	PJI of the hip	Biological: Phage therapy	A bacteriophage cocktail composed of phages BP13 and J1P3 is administered intra-articularly on day 1, followed by intravenous dosing twice daily from day 1 through day 14.	I/II
NCT06798168 [171]; Available	*Pseudomonas aeruginosa* MDR	Chronic PJI	Biological: Combining bacteriophage therapy with antibiotics for a case with hip PJI	Biological: Bacteriophage + Antibiotic Therapy Treatment includes weekly intra-articular injections of a personalized phage cocktail (*QDP-PSA-011*) for 3 consecutive weeks, combined with a 6-week course of antibiotics.	-
NCT06605651 [172]; Not yet recruiting	*Staphylococcus aureus*	PJI of the hip, Knee Prosthesis Infection	Biological: Anti-*Staphylococcus aureus* Bacteriophages (PP1493 and PP1815) intra-articular injection with 0.9% NaCl solution Drug: 0.9% NaCl solution	Experimental Arm (Active): Anti-*Staphylococcus aureus* bacteriophages (*PP1493* and *PP1815*) administered via intra-articular injection in a 0.9% NaCl solution. Placebo Comparator (Control Arm): Intra-articular injection of 0.9% NaCl solution only.	II

Abbreviations: PJI—prosthetic joint infection; DAIR—debridement, antibiotics, implant retention; NaCl—sodium chloride.

## Data Availability

This study is based on publicly available data retrieved from databases including PubMed, ClinicalTrials.gov, and Google Scholar. No new data were generated or analyzed by the authors.

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
