# Peer review of "Bacteriophage-Based Approach Against Biofilm Infections Associated with Medical Devices: A Narrative Review of ESKAPE Pathogens"

_ijms, 2025, doi:10.3390/ijms26178699_

Round 1

Reviewer 1 Report

Comments and Suggestions for Authors

The review titled "Bacteriophage-Based Approach Against Biofilm Infections Associated with Medical Devices: A Narrative Review of ESKAPE Pathogens" explores critical aspects of combating bacterial biofilms, a growing concern in healthcare. The rise in hospital-acquired infections and antibiotic-resistant bacteria presents significant clinical challenges. Biofilms represent a key survival tactic for bacteria in hostile environments, particularly on medical device surfaces. With biofilms implicated in infections and conventional treatments often proving ineffective, alternative solutions like bacteriophage therapy have gained attention.

This review delves into biofilm biology, focusing on the clinically relevant ESKAPE pathogens, and offers a thorough examination of how bacteriophages can serve as a novel strategy for biofilm management. Common biofilm-forming pathogens include Staphylococcus aureus, Pseudomonas aeruginosa, Klebsiella pneumoniae, Staphylococcus epidermidis, Enterococcus faecalis, Escherichia coli, Proteus mirabilis, and Acinetobacter baumannii. It investigates the mechanisms by which phages penetrate and dismantle biofilm structures, assesses existing therapeutic approaches leveraging these mechanisms, and discusses their limitations and future potential. This review consolidates findings from published research and ongoing clinical trials, highlighting the field's progress while underscoring the lack of large-scale, well-structured clinical studies.

The review is well written and summarizes a large amount of scientific information. I believe that this review can be published in International Journal of Molecular Sciences.

Comments:

  1. Line 70. Correct the typo “[14,15]Colonization”
  2. Figures 1 and 2 contain hidden text "Zawartość wygenerowana przez AI może być niepoprawna" and " AI-generated content ". Apparently, the authors used AI, but this is not described in the methods used.

Author Response

Dear R1,
We would like to warmly thank you for your thoughtful and constructive review of our manuscript “Bacteriophage-Based Approach Against Biofilm Infections Associated with Medical Devices: A Narrative Review of ESKAPE Pathogens.”

We are very grateful for your kind words about the clarity and scientific value of our work, and we truly appreciate your recognition that the review is suitable for publication in International Journal of Molecular Sciences. Your comments were precise, insightful, and have helped us improve the manuscript.

Comment 1:
 Line 70. Correct the typo “[14,15] Colonization”

Response 1:
 The correction has been made.

Comment 2:
Figures 1 and 2 contain hidden text "Zawartość wygenerowana przez AI może być niepoprawna" and "AI-generated content". Apparently, the authors used AI, but this is not described in the methods used.

Response 2:

The latest version of Microsoft Word automatically adds ‘alt text’ with AI-generated descriptions to any images, figures, or illustrations included in a document, accompanied by the disclaimer ‘AI-generated content may be incorrect.’ All figures in our manuscript were created entirely by the authors using the programs described in the methodology section, no AI tools were used in their preparation. The hidden alt texts have been removed.

All revisions in the manuscript have been highlighted using dark yellow font.

Thank you in advance!

Reviewer 2 Report

Comments and Suggestions for Authors

Following are my comments and suggestions for the editor’s decision:

  1. Table 1: It would be better to place the references in a separate column. However, much of the information here is already well-documented. What are the new insights provided by this table? I recommend preparing a comparative table highlighting EPS characteristics across all ESKAPE pathogens, since this aligns with the main theme of the paper.

  2. Figure 1: What structural differences exist in the biofilm mechanisms of each ESKAPE pathogen? Without such comparative details, the figure seems repetitive and similar to many published elsewhere.

  3. Table 2: The authors should present a comparative analysis of risk factors associated with ESKAPE pathogens.

  4. Table 4: Please separate the references into a dedicated column. Otherwise, the table is acceptable.

  5. Table 5: Instead of a simple list, this should be presented as a statistical figure summarizing research and review papers published between 2020 and 2025.

  6. Figure 2: This figure should illustrate the mechanisms of biofilm inhibition specifically against ESKAPE pathogens.

  7. Table 6: The authors should include details on concentrations and enzyme units of phages and phage-derived enzymes used for biofilm inhibition.

  8. Figure 4: This figure should be placed earlier in the manuscript.

  9. Limitations: The manuscript should include a separate section addressing the limitations and implications of phage-derived antibiofilm strategies.

Author Response

Dear R2, 

We would like to express our sincere gratitude for your exceptionally thoughtful and constructive review of our manuscript “Bacteriophage-Based Approach Against Biofilm Infections Associated with Medical Devices: A Narrative Review of ESKAPE Pathogens.”

Your insightful remarks and generous recognition of the scientific value and clarity of our work mean a great deal to us. It is truly an honor to receive such encouraging feedback from an expert with such a deep understanding of the subject.

Following your suggestions, we have undertaken a revision of the manuscript to improve its overall clarity and flow. Specifically, we reorganized the structure of the text to enhance readability, ensuring that the sections follow a more logical progression. In addition, we carefully revised the figures and tables so that they now appear in a coherent and consistent order, reinforcing the message of the manuscript.

We are convinced that these improvements have significantly strengthened the paper, and we owe much of this progress to your thoughtful guidance.

Responses to the Reviewer’s Comments:

Table 1: It would be better to place the references in a separate column. However, much of the information here is already well-documented. What are the new insights provided by this table? I recommend preparing a comparative table highlighting EPS characteristics across all ESKAPE pathogens, since this aligns with the main theme of the paper.

  • A comparative table highlighting EPS characteristics across all ESKAPE pathogens has been incorporated as suggested. In addition, the references have been placed in a separate column.

Figure 1: What structural differences exist in the biofilm mechanisms of each ESKAPE pathogen? Without such comparative details, the figure seems repetitive and similar to many published elsewhere.

  • The figure has been revised to include comparative details on the structural differences in the biofilm mechanisms of ESKAPE pathogens.

Table 2: The authors should present a comparative analysis of risk factors associated with ESKAPE pathogens.

  • The requested changes have been implemented and the table now includes a comparative analysis of risk factors associated with ESKAPE pathogens.

Table 4: Please separate the references into a dedicated column. Otherwise, the table is acceptable.

  • The references have been separated into a dedicated row, as recommended.

Table 5: Instead of a simple list, this should be presented as a statistical figure summarizing research and review papers published between 2020 and 2025.

  • We decided to retain the table format in order to ensure maximum clarity and accessibility of the information. By keeping the table, readers can easily identify the scope, type, and year of each study at a glance. We believe that in this case the tabular presentation provides a clearer and more practical overview, which is why we did not convert it into a statistical figure.

Figure 2: This figure should illustrate the mechanisms of biofilm inhibition specifically against ESKAPE pathogens.

  • Figure 2 illustrates the introductory text on bacteriophage therapy. It is not limited to the ESKAPE pathogens but rather depicts the mechanisms of action of phage-derived enzymes, which can be genetically engineered to target a wide range of microorganisms. The row titled “Spectrum” in the table part of this infographic has been removed, to avoid potential confusion for the reader. We believe this decision is appropriate, particularly after the reorganization of the manuscript structure, which now includes a new paragraph (Section 3.3). This section correlates the clinical use of phages with the ESKAPE group, following the preceding in-depth description of phage therapy approaches.

Table 6: The authors should include details on concentrations and enzyme units of phages and phage-derived enzymes used for biofilm inhibition.

  • Instead of incorporating these details into the summary table, we added information on concentrations and enzyme units directly in the text, highlighted in yellow, when citing the respective studies.

Figure 4: This figure should be placed earlier in the manuscript.

  • We have revised the order of the text, tables, and figures to improve clarity and readability, and Figure 4 has been repositioned accordingly.

Limitations: The manuscript should include a separate section addressing the limitations and implications of phage-derived antibiofilm strategies.

  • The limitations are discussed at the end of each phage-derived antibiofilm strategy section. We have also added new and additional details. This part now provides a clear structure that includes a description of the method, its advantages, and its limitations. While we did not create separate subsections, the presentation ensures clarity and readability.

All revisions in the manuscript have been highlighted using dark yellow font.

Thank you in advance!

Round 2

Reviewer 2 Report

Comments and Suggestions for Authors

Well done